# An Evaluation Method of Safe Driving for Senior Adults Using ECG Signals

**DOI:** 10.3390/s19122828

**Published:** 2019-06-25

**Authors:** Dong-Woo Koh, Sang-Goog Lee

**Affiliations:** Department of Media Engineering, Catholic University of Korea, 43 Jibong-ro, Bucheon-si, Gyeonggi-do 14662, Korea; metamo7@gmail.com

**Keywords:** safe driving intensity, driving anxiety, aggressive driving, cardiac response time, heart palpitations, electrocardiogram

## Abstract

The elderly are more susceptible to stress than younger people. In particular, heart palpitations are one of the causes of heart failure, which can lead to serious accidents. To prevent heart palpitations, we have devised the Safe Driving Intensity (SDI) and Cardiac Reaction Time (CRT) as new methods of estimating the correlations between effects on the driver’s heart and the movement of a vehicle. In SDI measurement, recommended acceleration value of vehicle for safe driving is inferred from the suggested correlation algorithm using machine learning. A higher SDI value than other people means less pressure on the heart. CRT is an estimated value of the occurring time of heart palpitations caused by stressful driving. In particular, it is proved by SDI that elderly subjects tend to overestimate their driving abilities in personal assessment questionnaires. Furthermore, we validated our SDI using other general statistical methods. When comparing the results using a *t*-test, we obtained reliable results for the equivalent variance. Our results can be used as a basis for evaluating elderly people’s driving ability, as well as allowing for the implementation of a personalized safe driving system for the elderly.

## 1. Introduction

The proportion of elderly individuals is increasing around the world, and the safety of vehicles that are operated by seniors has emerged as a topic of interest in the field of intelligent vehicles. Elderly drivers drive less frequently than their younger counterparts, and accidents involving seniors are generally attributed to age-related disability or physiological disadvantage, which increases the likelihood that elderly drivers and their passengers are killed or injured in a car accident [1,2]. Examples are abundant worldwide: in Chicago, USA, a reckless 81-year-old driver caused a major accident in which three people were killed and 20 were injured [3]; in Daejeon, South Korea, a 63-year-old taxi driver had a heart attack while driving and his car only came to a halt after crashing into another vehicle [4]; in British Columbia, Canada, a 57-year-old driver had a heart attack while driving and crashed into two parked cars and a pickup truck [5]. Such accidents reflect the frailty of seniors and the considerable risk of death associated with them driving. Driving stress has been shown to be highly related to anxiety and to have deleterious effects on driver behavior, which can result in traffic accidents [6]. Anxiety has also been shown to be highly related to aggressive driving, which is committing unprovoked attacks on other drivers, such as not yielding to vehicles wishing to pass [7]. Nearly eight out of every ten US drivers admit expressing anger, aggression, or road rage at least once in the previous year, according to a survey by the AAA Foundation for Traffic Safety [8]. Many older adults experience driving anxiety with quite a large proportion reporting high levels of driving anxiety and associated differences in driving patterns [9].

Since strong emotional reactions are known to release adrenaline [10], which can trigger adverse cardiac events related to the cardiovascular system [11], it is important to be able to quantify stressful driving situations with heart rate acceleration and prevent car accidents [12]. In particular, senior drivers typically exhibit impairment of perceptual, psychomotor, cognitive, and physical abilities [13]. The nature, extent, and manifestations of such impairments differ from person to person, depending on their individual aging status. Specifically, driving ability can vary greatly depending on the individual’s degree of aging [14]. To check exact driving status for elder safety, the movement of the vehicle and the driver’s condition should be checked. In particular, quantified values are needed to compare individual driving capabilities. This should model and quantify the relationship between the driver’s condition and the vehicle’s motion. Assessing the driving abilities of the elderly with disabilities is often a very challenging task because each medical condition is accompanied by physical impairments, and relative individual functional performance may vary depending on personal characteristics [15].

For this reason, most efforts to detect the aggression-related emotional status of drivers have focused on developing technology that recognizes potentially dangerous situations based on the driver’s facial expression [16]. However, facial recognition technology does not perform well under certain circumstances (e.g., if the person is wearing glasses, has an impassive face, or the lighting is poor). Therefore, detection methods of aggression-related emotions with measurable bio-signals such as with electrocardiograms (ECG) have been investigated. Additionally, the autonomic nervous system, which regulates the function of internal organs in response to various stimuli, can also provide information regarding stress levels [17] and impending cardiac events, which can then be used to develop warning systems, such as the advanced driver assistance system (ADAS).

In clinical applications, ECG signals are acquired using 12 or more leads that are mounted on the chest and limbs, using gel to improve the conductivity [18]. These complex medical devices can measure accurate ECG data, but they are inconvenient to use while driving. In recent years, some methods have been proposed to try to simplify ECG signals using a minimum number of leads [19], including performing an ECG with a portable gadget [20]. Here, we propose a lightweight and robust algorithm for R-Peak detection, a key factor in ECGs, for use in vehicles. Additionally, we present a novel model of the relationship between ECG and vehicle motion that accurately detects the driver’s state and the Safe Driving Intensity (SDI).

Our proposed methods make several key contributions. The first goal of this study is to quantitatively determine the changes in stress due to stressful driving situations between elderly and young people with a cardiac condition before driving and while driving. People with a longer driving history tend to overestimate their driving abilities [21], so they are expected to provide low-stress responses to the questionnaire on driving. In assessing these changes in stress, we can evidently compare the results of the self-evaluation questionnaires with the analysis of stress sensitivity measured while driving, as indicated by an electrocardiogram. The second goal is to quantify and model aggressive vehicle movements that are related to the driver’s heart palpitations via stressful driving situations. Such a model is presented as the individual Safe Driving Intensity (SDI) that is used while monitoring heart rate acceleration changes due to aggressive vehicle movement. We also calculated the Cardiac Response Time (CRT) to predict the timing of heart palpitations that may occur with high-intensity vehicle movement, such as stressful driving situations. In doing so, we present a novel model of the relationship between ECG and vehicle motion that accurately detects the driver’s state based on SDI and CRT.

The paper is organized into five subsequent sections. Section 2 presents related studies that have assessed cardiac stress levels during stressful driving. Section 3 provides an overview of our proposed system and describes the experimental set-up involving ECG measurements during a session of stressful driving. Section 4 further demonstrates the procedure used to obtain the SDI with a simple R-Peak detection method. Section 5 discusses the results for young and elderly drivers. Finally, Section 6 summarizes the results of our study and their implications.

## 2. Related Work

A considerable amount of research has been conducted to develop ways to detect threats during driving, with the main foci being on stress [17,22,23,24,25], fatigue [26,27,28], and aggressiveness [29,30,31,32,33]. These factors are linked to stressful driving situations and are associated with increased risk of fatal accidents [6]. Unfortunately, elderly drivers are very sensitive to stress, fatigue, and aggressiveness during driving due to the deterioration of their physical ability, but there are few related studies. Although a variety of studies have been conducted to identify driver’s conditions causing driving stress, they could not present quantified values universally. Machot et al. [34] proposed a method based on speech recognition to assess the driver’s mood. However, speech recognition is not appropriate in vehicles with a noisy environment. Others suggested mood indicators including changes in ECG parameters, skin temperature, and skin conductivity. Among these methods, ECG time-domain information involves emotional state in the distance between R-Peaks. Kim et al. [35] and Healey et al. [22] examined the correlation between various biometric parameters and emotional states. Picard et al. [36] suggested an intelligent system for automatic detection of emotions, which combines information from several biometric signals, such as ECGs and electromyograms.

Healey et al. [22] applied a wide range of detection methods (e.g., ECG, electromyography, skin conductivity measurements) in an effort to determine the emotional state of the individual, but their approach cannot be applied for assessing emotional status while driving, as it involves the use of many devices that would distract the driver. In our research, a small portable ECG gadget was used to comfortably check the driver’s condition and, as seen in previous studies, we found that an ECG provides sufficient information to analyze emotional and physical states.

In a vehicle, an ECG device can be used to detect changes in an individual’s physical state, such as fatigue [26,27] and drowsiness. Additionally, ECG signals have been shown to closely reflect changes in the emotional state [37]. Lee et al. [23] sought to detect driving stress in people driving on public roadways, highways, and within the city center. The analysis of the ECG data revealed that drivers had higher heart rates when driving on busy, narrow streets with many shops. Hayano et al. [38] compared a driver’s heart rate to their estimated manipulations, such as accelerating, and found that it corresponded to the vehicle’s speed. These results suggest that heart rate is a strong indicator of physical state in situations on the road [24]. Healey et al. [22] analyzed data from 24 driving sessions that were at least 50 min in length. Their model used the standard deviation of normal RR intervals (SDRR) calculated over 5-min intervals obtained from individuals during periods of rest while driving on a highway and in the city. In doing so, they were able to distinguish between three levels of stress with an accuracy of over 97% across multiple drivers and driving days. However, the upper SDRR over 5-min intervals is not adaptable to accurately estimate a driver’s state during situations of fast driving. Therefore, we used 30-s intervals for detecting a tensioned ECG period with standard deviation of RR intervals.

An estimation of the driver’s exact stress level suggests that SDRR is a stable indicator of stress and can be extracted relatively easily from ECG data. Munla et al. [17] also collected ECG data that reflected heart rate variability and found that a classifier based on support vector machines can improve stress detection capabilities. Changes in ECG signals (heart rate variability) are typically characterized in terms of the peak-to-peak (RR) interval [39], and heart rate variability [40] has frequently been studied as a general indicator of stress. Time-domain signals of an ECG are considered useful for describing emotional features [41,42]. Friedman et al. [43,44] described the manifestation of stress as a feeling of anxiety followed by physical transformation (increased heart rate) with behavioral adaptation for recognizing signs of danger (hypervigilance). Rapid heart rate can interfere with normal heart function and increase the risk of sudden cardiac arrest. Heart rate variability represents the physiological phenomenon of the variation in the time interval between heartbeats [45]. To analyze heart rate variability, the R-Peaks must first be detected so that the RR interval can be calculated. Although the RR interval is the strongest indicator of stress while driving, this parameter also depends on the characteristics of each driver (e.g., age, physical condition). Moreover, driving propensity is greatly affected by the psychological condition of each driver [46]. For these reasons, it is important to assess the cardiac stress threshold associated with a diminished driving ability in each driver.

Most recent studies have focused only on the state of the driver and do not consider the relationship between the physical state of the driver and that of the vehicle, such as acceleration and sliding, for estimating a safe driving speed. In several studies focused on stressful driving, movement was assessed using an acceleration sensor [29,30,31,47], but excluded the driver’s state. To address this limitation, our research attempts to model the physical and mental condition of the driver, which is reflected in the ECG signal, while accounting for the state of the vehicle, which is reflected in its movement parameters of acceleration. The SDI proposed in this study can quantify the degree of stress due to stressful driving situations. In particular, we suggest that the CRT measured during stressful driving can be used to predict abnormal heart palpitations in the driver. In doing so, we demonstrate heart palpitation detection or an early warning system for safe driving.

## 3. Overview

Figure 1 shows the block diagram of our proposed SDI detection system. During data acquisition, ECG (RA-LL) and Accelerometer X data are acquired from young and elderly people using mobile ECG equipment (Shimmer3 EEG unit; Shimmer, Dublin, Ireland). All results of Shimmer3 were measured in the vehicle (Carnival; Kia, Korea). During R-Peak detection, ECG data are normalized. R-Peaks and RR intervals are then extracted using a robust suggested statistical algorithm.The RR interval means the time between two subsequent heartbeats. Many researchers used the RR interval because it contains a lot of information such as emotional state, physical condition, and so on. Other studies focused on fatigue and stress during driving. However, our study focused on heart palpitations caused by stressful driving. Heart palpitations are a medical term but very subjective. Therefore, unsupervised learning is used to analyze detecting heart palpitations, including the status of each individual.

Heart palpitations can be caused by various psychological or physical reasons. The main point is that heart palpitations are a major condition of the body, so it is important to prevent heart palpitations to avoid serious heart-related accidents during driving. Therefore, this study proposed that SDI and CRT can be used in preventing and estimating heart palpitations. The SDI can provide a quantitative measurement of how stressful driving affects the heart of driver for driving assessment, and can be used to predict the occurring threshold of heart palpitations that is acceptable for safe driving.

To determine our SDI, measured RR intervals are passed into K-means clustering, after which high-tensioned RR interval data are derived. We assume that heart palpitations are the high-tensioned RR interval data obtained through k-means clustering, representative of unsupervised learning, and extracting acceleration sensor data indicative of heart palpitations. SDI is determined among these extracted acceleration sensor data by the proposed method with the overlapped sliding time window. In short, a high SDI means that the driver is prone to not experience heart palpitation. Additionally, the averaged time between the vehicle’s acceleration and heart palpitations RR intervals is defined as the CRT. Finally, the differences in ECGs between young and old drivers were analyzed.

The SDI and CRT, designed with individual driver status, showed an important quantitative indicator for stable driving operation. Furthermore, the SDI was validated by other general statistical methods(GV, gradient validation). In particular, we proved that elderly drivers have lower SDI results than young drivers in stressful driving situation even though elderly subjects overestimated their driving abilities in the personal assessment questionnaire for long experience.

## 4. Methods

We expected that younger and elderly drivers would have different cardiac statuses during driving. It is hypothesized that elderly drivers would react more sensitively during the stressful driving situation. For this reason, both elderly and young adults were recruited.

All participants were licensed and active drivers. Twenty-four male young and older adults were recruited. However, data from one young and one elderly subject had been incorrectly logged and they were excluded from the analysis. We therefore performed our analysis on data from 22 male subjects (Table 1), consisting of 11 elderly drivers (11 subjects between 61 and 72 years; M = 65 years, SD = 3) and young drivers (11 subjects between 22 and 35; M = 27 years, SD = 6).

Participants were first informed about their task. Prior to the driving test, participants who signed an informed consent form filled out a questionnaire. All subjects practiced on the driving course and, once accustomed to it, began driving. The questionnaire was composed of three parts. Parts 1 and 2 were completed before the experiment and consisted of questions listed elsewhere [48,49]. Part 1 [48] consisted of a self-assessment of driving skill, where higher scores were associated with an overestimated perception of driving ability. Part 2 [49] quantified aggressive tendencies, with a high score indicating higher susceptibility to stressful driving. Part 3 was completed immediately after the experiment and consisted of simple questions assessing the driver’s level of concentration and perceived performance.

To measure various cardiac responses in the real driving test, including strong response, the experimental course was composed of varied-difficulty courses: straight, sharp turn, and sliding. For each subject, the total duration of the experiment was 11 min, and the data collection period was divided into three stages: before driving, during driving, and after driving (Figure 2). During the first minute after initiating the ECG measurement, the subject is instructed to relax. After completing the driving stage, the subject is also instructed to relax. The collected data points for the first 500 ms were excluded from the analysis because more external noise was generated for the first 500 ms of the experiment due to the characteristics of the equipment.

Two operators were aboard during each driving test. One operator verified that the ECG was working, and the other operator instructed the subject to increase driving intensity if he did not drive aggressively. These demands were to keep the driver constantly stressed and to induce aggression. The Shimmer3 is a compact mobile device that is simply mounted on the subject’s skin. A total of five leads (RA, LA, LL, RL, V2) were attached, but only RA-LL data were used in the experimental analysis. The built-in accelerometer of the Shimmer3 was used to correctly synchronize with the ECG.

### 4.1. R-Peak Detection

The QRS waveform of the elderly drivers was weaker than that of the younger drivers (Figure 3). This made R-Peak detection difficult. Additionally, mobile devices in the vehicle offer low-quality signals compared to medical applications for driving situation. To detect these signals, we proposed a simple and robust new method. This method detects the QRS discovery unit, taking into consideration the age of the subject. First, ECG data normalization for time series using sliding window was intensively performed for a short segment of time. This is because the ECG with the vital voltage level changes in phase dramatically according to a subject’s body status and cannot be analyzed without normalization. Second, after normalization, our study suggested a simple R-Peak detection method suitable for a mobile environment. For R-Peak detection, the values below the sum of the mean and the standard deviation are removed. To detect R-Peak, we constructed a time window frame that considered the age of the driver. The maximum value within the time window frame was defined by the R-Peak value. R-Peaks detected within too short an interval—the maximum heart rate [50]—were regarded as abnormal data and were removed for noise canceling. All subjects’ ECG data had noises for driving situation. Additionally, the data of elderly people was not particularly good compared with young people. For this reason, our R-Peak detection method is suggested and specific to real-time portable devices in a noisy environment. The detailed detection method is as follows.

Normalizing ECG data over shorter segments of time allowed us to eliminate excessive fluctuation of the waveform with consideration of the biometric level. Time normalization for the time series of ECG using sliding window can be expressed as follows:(1)$$g_{i}\left(t\right)\equiv \frac{G_{i}\left(t\right)-\langle G_{i}\rangle }{\sigma_{i}}$$
where $\sigma_{i}$
$G_{i}$, and $\langle ...\rangle$ denotes a time average over the period. $G_{i}\left(t\right)$ denotes the value of original ECG data. $g_{i}\left(t\right)$ is a normalized value of the original ECG data by Equation (Equation 1). The signals are normalized with respect to 0 mV and the resulting R-Peak can be detected using our proposed method. First, the expected R-Peak is chosen as the value over the sum of the mean value and standard deviation, since the R-Peak is defined as the greatest value in a QRS wave [45]. It is based on the mean value in each of the three stages of the experiment (before, during, and after driving). Additionally, Time Window size for R-Peak Detection (TWRD) is also required to determine the real R-Peak value among candidate data. It is decided based on the pulse rate during exercise depending on the individual’s age. Also, the size of the time window is determined from the premise that a proper exercise heart rate cannot occur during driving referring to the vigorous exercise intensity heart rate [50]. Under this assumption, it is also assumed that an R-Peak of only 0 or 1 can be generated from the data within TWRD. If the data is detected within this range, then the maximum value can be a candidate for the R-Peak. If the R-Peak data is detected incorrectly, it would be detected in duplicate R-Peaks within another time frame: the maximum heart rate [50] which absolutely cannot be generated during driving. In this case, the duplicated later signal was deleted. The time required to filter noise from the maximum pulse rate is determined from the subject’s age. It should be noted that these procedures are specialized for mobile vehicle systems and not for more elaborate medical purposes. Our proposed scheme for R-Peak detection enables us to assess cardiac stress levels according to driving strength in a noisy vehicle environment (see Section 5).

### 4.2. Calculation of SDI and CRT

Our research focused on finding the correlation between driver’s ECG parameters and the acceleration of the vehicle. We sought to identify and quantify the critical point of safe driving strength to prevent heart palpitations, defined as SDI. To obtain the SDI, the K-means clustering technique was used. The RR intervals were measured, and K-means clustering was used to detect tense moments during driving. The relationship between a tense moment in the ECG and vehicle acceleration was modeled to determine the SDI, which was confirmed with a simple gradient method. K-means clustering is defined as
(2)$$\underset{S}{\mathrm{argmin}}\sum_{s=1}^{k}\sum_{i=1}^{n}\left|\right|X_{i}^{\left(s\right)}-\mu_{i}{\left|\right|}^{2}$$
where *S* denotes objective function {$S_{1},S_{2},...,S_{k}$}, *k* denotes the number of clusters, *n* denotes the number of cases–RR distance values, *x* denotes the RR distance, and $\mu_{i}$ is the mean of points in $S_{i}$. Finding the minimum value $S_{i}$ is the goal of the algorithm. The algorithm starts by setting the initial $\mu_{i}$. Then the next two steps are Assignment step Equation (Equation 3) and Update step Equation (Equation 4). If the cluster does not change, it stops repeating.
(3)$$S_{i}^{\left(t\right)}=\left\{X_{p}:|X_{p}-\mu_{i}^{\left(t\right)}{|}^{2}\leq {\left|X_{p}-\mu_{j}t\right|}^{2}\forall j,i\leq j\leq k\right\}$$
(4)$$\mu_{i}^{t+1}=\frac{1}{|S_{i}^{\left(t\right)}|}\sum_{x_{j}\in S_{i}^{\left(t\right)}}x_{j}$$

However, we do not use all the clusters calculated by K-means clustering above. The cluster group value below the sum of “average and standard deviation” of clustered representative values is not regarded as tense RR interval data as in Equation (Equation 5).
(5)$$s_{i}=\left\{x_{i}:|x_{i}|>=\frac{1}{n}\sum_{i=1}^{n}\left|\mu_{i}+\sigma_{i}\right|\forall i,1<=i<=k\right\}$$

Above, $s_{i}$ value is regarded as heart palpitations ECG data during stressful driving and mapped to the acceleration sensor value for finding the suggesting SDI, CRT. It is difficult to find exact matching between caused acceleration data and effected ECG data. Therefore, the caused acceleration data was selected from the preceding time of heart palpitations ECG data (see Figure 4).

It is assumed that the heart’s reaction due to strong vehicle movement will react differently depending on many factors, such as the subject’s age. Therefore, our study focused on extracting the relation model with SDI/CRT between stressful driving and the heart’s reaction. Above all, the SDI has been devised to determine how strong driving can lead to mainly stressful cardiac conditions. From the point of occurrence of the high tense ECG data of Equation (Equation 5), the maximum acceleration value within the previous MTW magnitude is found, and the average value of the found values becomes SDI. The reaction time is also obtained from the time between maximum acceleration value within the previous maximum window size and heart palpitations ECG data. They were summed, averaged, and defined as CRT. Thus, SDI/CRT can be derived as Algorithm 1 (see Figure 5). It also shows CRT based on the extracting method of SDI (see Figure 6).

SDI is a completely new concept for the evaluation of safe driving, so it needs to present authentic results. For that reason, we attempted to verify the SDI based on the amount the acceleration value during stressful driving in the opposite direction to the suggested method of SDI. We assumed the range of acceleration values that can occur during driving, and then changed the value in unit variable. In addition, it measured the number of heart palpitation RR intervals according to the unit variable (0.1 m/s${}^{2}$) to get a histogram. The number of the heart palpitation RR interval was counted every 0.1 m/s${}^{2}$ (see Figure 7), changing from 1.0 to 6.5 m/s${}^{2}$. If the highest gradient values were generated during high-intensity driving, then that point was defined as the stressful driving period. Accordingly, the acceleration value with the highest gradient might represent the safe driving threshold value, which is used to validate the SDI with the *p*-value of a *t*-test for safe driving. In short, this method is simple and faster than the previous methods. However, if there is no obvious gap or there are several similar gaps between gradients, this method may not accurately indicate the threshold for safe driving ability. This procedure should only be used as a complementary method for validation.

## 5. Results

### 5.1. Questionnaire Results

Table 2 shows the questionnaire results. Unlike younger people, elderly people scored their driving ability highly in parts 1 and 2, whereas elderly people assessed their driving concentration (part 3) as being lower than younger people. However, the results were different between the RR intervals obtained during the driving experiment and the self-evaluation on the questionnaire. Elderly people were found to be less nervous than younger drivers, based on the questionnaire data. However, there was a larger difference between the before and during driving results in senior adults than in younger adults in the actual driving experiment. Thus, elderly people may be overly confident about their ability to drive for long periods of time.

### 5.2. Comparing Averaged RR intervals Time between Driving and Not Driving

The elderly (>60 years old) have more cardiac variation than younger adults (<30 years) between driving and not driving (Table 3). This greater variation results in higher tension among the elderly, which conflicts with their result in part 3 of the questionnaire. On the other hand, there was no significant difference between young adults and elderly adults in the before-driving and after-driving stages. These results clearly demonstrate that the RR interval can discriminate between driving and not driving depending on the age group. In general, young adults (such as the 26-year-old in this experiment) had sharp, strong QRS R-Peak waves during driving. In contrast, data for seniors (such as the 72-year-old in this experiment) showed imprecise QRS waves and unstable fluctuation of the R-Peak during driving (see Figure 3). In conclusion, these results show that the R-Peak detection method proposed in this study can be obtained by very effective operation to detect the ECG data of elderly people who have lower QRS waves.

### 5.3. Determining SDI and CRT

This study generally assumed that a stressful driving period can occur when acceleration changes quickly. It focused on determining the SDI by the correlation point between heart palpitation ECG data and stressful driving acceleration data, which must be a boundary to determine safe driving. If a subject has a high SDI value, it means that he has a good ability to drive through unpredictable obstacles, such as vehicle sliding, with an appropriate cardiac load. This feature is thus useful for differentiating safe driving assessment. The CRT can predict the heart response time to an unexpected obstacle. If the CRT is short, it means that the heart reaction occurs quickly. They are expected to be useful as a reference when configuring a safe driving system preventing heart palpitations. The SDIs and CRTs for all subjects are provided in Table 4. Notably, the SDI value was reliable (0.80 p) for the equivalent variance using a *t*-test with results obtained during gradient validation(GV, Figure 7). Additionally, the R value of PCC (Pearson’s Correlation Coefficient) also exhibited a strong relationship, as detailed in Table 5.

## 6. Conclusions 

In this paper, we introduced the SDI, which successfully models the relationship between the vehicle’s acceleration and the driver’s heart rate. The CRT was also performed to predict the driver’s heart palpitation state. Our proposed R-Peak detection method demonstrated that the ECG R-Peaks were successfully detected in a noisy environment using the suggested simple statistical method. It operated well with the low amplitude ECG data of some elderly. The RR interval analysis showed that the heart rate of the elderly had more changes during driving on average compared with young people. However, the changes of heart rate alone could not explain the condition of safe driving precisely. The SDI enabled the adaptation of quantitative estimation for preventing heart palpitations. The CRT was also able to predict the time of excessive heart palpitations caused by stressful driving. As a result of the comparison analysis with the SDI and CRT, we found that elderly people were more prone to exhibit heart palpitations during stressful driving, generally. However, the difference is rather small. We need more research on individual conditions considering the individual’s psychological and physical conditions. Unfortunately, the relationship between the CRT and SDI has not been clarified in this study. Nevertheless, our results effectively detected the occurring threshold of heart palpitations, and can use this for evaluating elderly people’s driving ability. In the future, gathering additional ECG data for various roads and longer driving sessions, as well as combining the methods described herein with a prediction algorithm, would allow for the implementation of a personalized safe driving system preventing serious car accidents related to heart rate that proactively could manage potential risk.

## Figures and Tables

**Figure 1 sensors-19-02828-f001:**
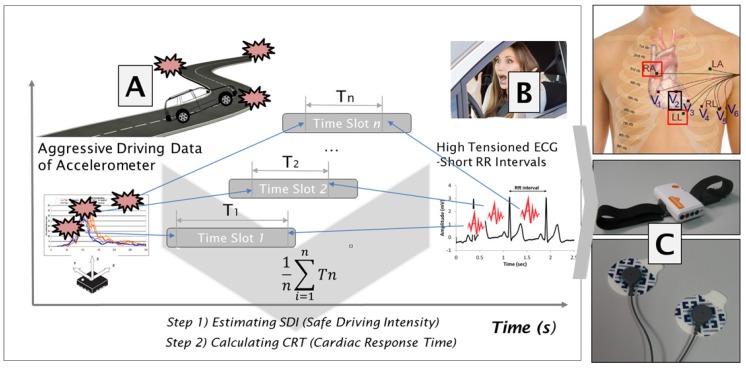
Illustration of the SDI and CRT, which are correlated interactions between (**A**) and (**B**). It is an important measure to evaluate safe driving ability. The portable Shimmer3 EEG unit(Shimmer, Dublin, Ireland) was used to monitor the ECG signal while driving (**C**).

**Figure 2 sensors-19-02828-f002:**
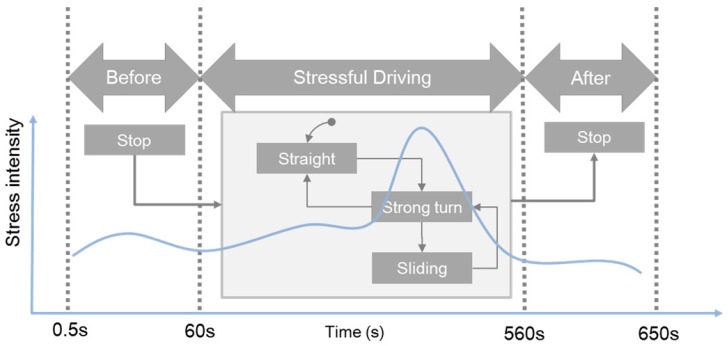
Experimental stages. Data were collected over the course of 11 minutes divided into three stages. During the main stage of the experiment (stressful driving; 500 seconds), the subjects were instructed to drive a vehicle on a difficult course with sharp turns. During the initial (before-driving) and final (after-driving) stages of the experiment, the subjects were instructed to relax.

**Figure 3 sensors-19-02828-f003:**
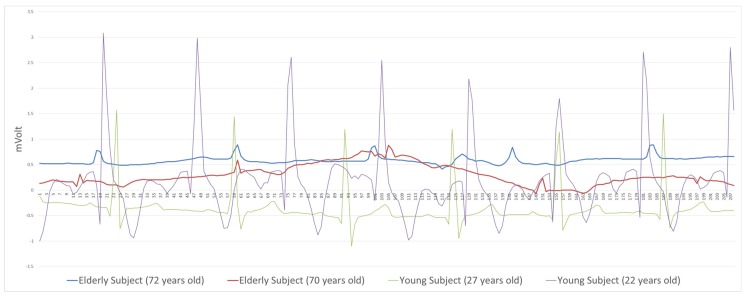
Sample ECG original signals that exhibit big differences between young and elderly subjects. The QRS amplitude of elderly people tends to be weaker than that of young people.

**Figure 4 sensors-19-02828-f004:**
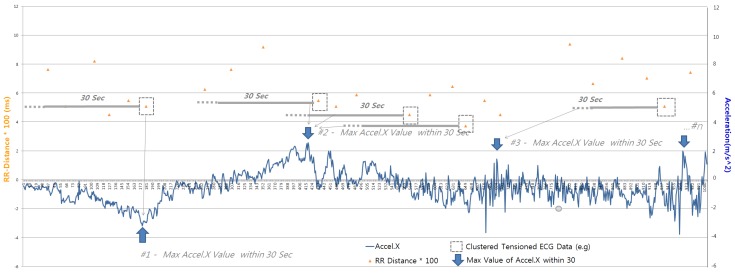
#n points of maximum acceleration within MTW (30 s, Maximum Time Window). These moments were considered to be most stressful driving moment, which cause heart palpitations.

**Figure 5 sensors-19-02828-f005:**
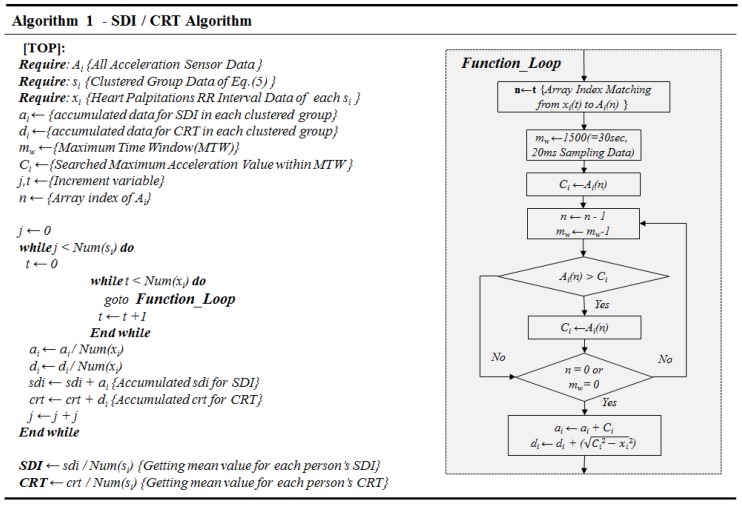
Main algorithm of SDI/CRT.

**Figure 6 sensors-19-02828-f006:**
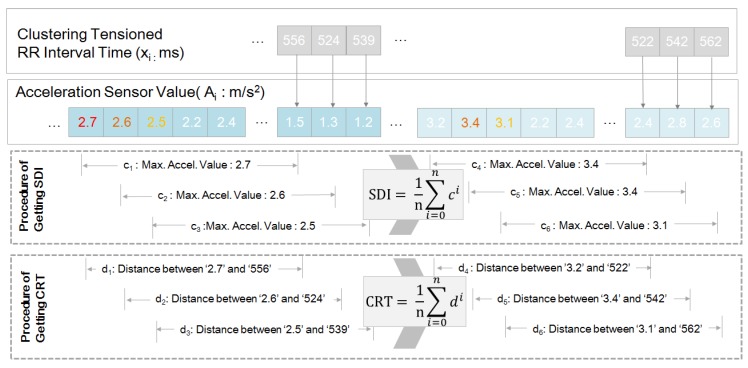
Procedure for obtaining the SDI and CRT. As shown here, the SDI is originated from the acceleration value and the CRT is the averaged distance between maximum the acceleration value within the MTW and each clustering heart palpitations RR interval value.

**Figure 7 sensors-19-02828-f007:**
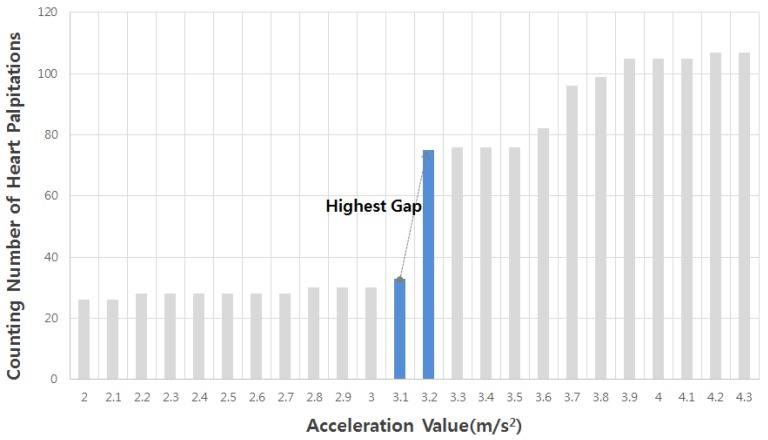
Histogram of the heart palpitation summation at every acceleration value per 0.1 m/s${}^{2}$. As shown here, the safe driving intensity value is 3.1 according to the highest gap obtained using this method.

**Table 1 sensors-19-02828-t001:** Characteristics of the Young and Elderly groups.

Variable	Elderly Group *n* = 11	Young Group *n* = 11	T Statistic	*p*-Value
	M (SD)	M (SD)		
Demographics Age (years)	65.18 (3.43)	26.55 (3.26)	27.29	<0.0001
Driving characteristics Experience (years)	39.91 (4.44)	5.55 (3.26)	20.35	<0.0001

**Table 2 sensors-19-02828-t002:** Questionnaire Results.

Subject Group	Questionnaire					Tstatistic	*p*-Value
	Part	Target	Total Score	M	SD		
Y		Skill	50	19.45	7.35	0.52	0.61
E	1			18.00	5.62		
Y		Aggressiveness	85	39.00	10.58	0.57	0.57
E	2			36.18	12.50		
Y		Concentration	35	16.09	4.53	4.42	<0.001
E	3			9.09	2.66		

**Table 3 sensors-19-02828-t003:** The greatest difference in RR intervals in elderly drivers was observed between during driving and before driving. This gap was larger than that in young drivers. Generally, the RR interval becomes longer as drivers get older.

Driving Stage	Subject Group	RR Interval (ms)		T Statistic	*p*-Value
		M	SD		
BEFORE	Y	649.43	76.20	−5.3	<0.0001
	E	812.29	63.18		
DRIVING	Y	624.63	82.49	−3.92	<0.0001
	E	752.87	65.66		
AFTER	Y	657.93	85.74	−4.07	<0.0001
	E	803.36	77.32		

**Table 4 sensors-19-02828-t004:** SDI/CRT for each of the 22 subjects.

Method	SDI (ms${}^{2}$)				CRT (s)			
Subject	Elderly Group	Youth Group	T Statistic	*p* Value	Elderly Group	Youth Group	T Statistic	*p* Value
1	3.75	3.18	−0.44	0.66	13.79	13.52	0.69	0.50
2	3.37	3.60			13.94	13.89		
3	3.96	3.71			14.83	12.9		
4	3.35	3.41			14.88	13.97		
5	2.99	3.23			13.96	14.49		
6	3.29	2.28			14.03	15.35		
7	2.79	3.45			12.71	13.26		
8	2.49	3.69			14.81	13.76		
9	2.57	3.29			13.19	14.4		
10	4.04	2.97			14.71	7.58		
11	2.90	3.69			13.54	16		
M	3.23	3.32			14.035	13.556		
SD	0.53	0.42			0.7205	2.1756		

**Table 5 sensors-19-02828-t005:** SDI Validation Result Using the Gradient Method.

Method	T Statistic	*p* Value	PCC	M	SD
SDI	0.26	0.80	0.61	3.27	0.47
GV				3.24	0.47

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
