# Peer review of "An Evaluation Method of Safe Driving for Senior Adults Using ECG Signals"

_sensors, 2019, doi:10.3390/s19122828_

Round 1

Reviewer 1 Report

1.     The abstract is an important section of a paper. Its main goal is to present in a clear and concise way, the central idea of the work and its novelty. The authors must improve the abstract section in this sense. They must focus their explanations.

2. It would be interesting that the authors motivate that the results obtained and presented in this paper are limited to application of specific characteristics explained.

3.     We suggest that the conclusion section must be more focused. The first paragraph is interesting but long a general for this kind of section. It would be necessary to improve a bit this section

Author Response

(Writer’s Answer)

A.   I really appreciate your detailed review and suggestions. Thank you very much.

1.     The abstract is an important section of a paper. Its main goal is to present in a clear and concise way, the central idea of the work and its novelty. The authors must improve the abstract section in this sense. They must focus their explanations.

(Writer’s Answer of 1)

A.   Almost all of them were modified according to your suggestions. Thanks.

2. It would be interesting that the authors motivate that the results obtained and presented in this paper are limited to application of specific characteristics explained.

(Writer’s Answer of 2)

A.   Thanks.
Additionally, heart palpitations can occur due to various psychological or physical causes. The main point is that heart palpations can be a major problem, so it is important to prevent them before a serious heart-related incident occurs. Therefore, this study proposed using the SDI and CRT to prevent heart palpitations with suggested robust R-Peak detection algorithm in a nosy environment (Especially, our R-Peak detection algorithm operated well with the low amplitude ECG data of some elderly). Our results can be used to detect the threshold of heart palpitations that is acceptable for safe driving and can use the data as a basis for evaluating elderly people's driving ability.

3.     We suggest that the conclusion section must be more focused. The first paragraph is interesting but long a general for this kind of section. It would be necessary to improve a bit this section

(Writer’s Answer of 3)

A.   I modified conclusions according to your opinion, and corrected and improved the inaccuracies, additionally.

My paper was improved so much for your very helpful feedback

Thank you once again.

Best regards.

Reviewer 2 Report

Nice paper and idea. Important in understanding driving capability for elders.

Some comments:

- Fig. 4 Not clear - axes description requires

- As for Fig.3 - ECG amplitude may vary from person - to person.

- Regarding R-peak detection, there is Pan-Tompkins algorithm, well described and applicable bith, in hardware and software. Authors propose own solution.

- In conclusions: SDI models relationship between vehicle's acceleration and driver's electrocardiogram. In my opinion this statement is too strong. Electrocardiogram is much more than heart-rate.

- in general - Fourier transform should not be used for R-R period detection. A time-frequency based spectrograms can be applied, but in general - just refer to Pan-Tompkins.

- If ECG is used only for HR estimation, wouldn't be better to use photopletysmography instead ? Less noise, easier to calculate, applicable on finger or earlobe

- What was delay shift between excessive acceleration moments and increased R-R frequency ? One might expect different time shift among evaluated persons ?

- Detected R-R events are not equally sampled date. Despite 50Hz Shimmer sampling, detected R-R values occurs in different time moments. In order to compare results in some objective method , detected R-R intervals should be interpolated (e.g spline) and resampled. This allow to be compared with measured acceleration energy - which should be taken as modulus of acceleration.

- suggestion - in future research gather OBD signals from the car - accelerations and brake signals can be observed. Intensity of acceleration and frequency of braking can support information about driver's mud.

Author Response

Nice paper and idea. Important in understanding driving capability for elders.

(Writer’s Answer)

A.   I really appreciate your detailed review and suggestions. Thank you very much.

Some comments:

[1] - Fig. 4 Not clear - axes description requires

(Writer’s Answer of 1)

A.   I drew a line and added the caption again. The left axis means RR-Intervals. The right axis means the acceleration value.

[2] - As for Fig.3 - ECG amplitude may vary from person - to person.

(Writer’s Answer of 2)

A.   In the Shimmer 3, elders ECGs had been measured with a lower amplitude compared to the data of young people. In the experiment, two professional operators riding in the vehicles did double check, and it was not the equipment's fault. To overcome this problem, we proposed the new R-Peak detection method in this study.

[3] - Regarding R-peak detection, there is Pan-Tompkins algorithm, well described and applicable bith, in hardware and software. Authors propose own solution.

(Writer’s Answer of 3)

A.   To overcome the above problems (refer to “Writer’s Answer of 2”), I had mainly checked only the most popular algorithms using Fourier transforms. After checking Tomkins' algorithm per your suggestion, I thought it was a very good algorithm using traditional signal processing techniques.

 [4] - In conclusions: SDI models relationship between vehicle's acceleration and driver's electrocardiogram. In my opinion this statement is too strong. Electrocardiogram is much more than heart-rate.

(Writer’s Answer of 4)

A.   I modified conclusions according to your opinion, and corrected and improved the inaccuracies, additionally.

[5] - in general - Fourier transform should not be used for R-R period detection. A time-frequency based spectrograms can be applied, but in general - just refer to Pan-Tompkins.

(Writer’s Answer of 5)

A.   I deleted the comments that said the time-frequency based spectrograms are special. And I also deleted the explanations comparing our algorithms to algorithms using Fourier transforms.

[6] - If ECG is used only for HR estimation, wouldn't be better to use photopletysmography instead ? Less noise, easier to calculate, applicable on finger or earlobe

(Writer’s Answer of 6)

A.   It's a good idea, if it can overcome the interference attaching finger during driving. At the time of the research plan, I intended to use the existing Shimmer 3 because of the budget problem. I would like to use it in future experiments.

[7] - What was delay shift between excessive acceleration moments and increased R-R frequency ? One might expect different time shift among evaluated persons ?

(Writer’s Answer of 7)

A.   The original plan was intended to give different delays depending on the subject’s characteristics, but it was a very difficult task. As a result, we used a fixed delay time of 30 seconds.

[8] - Detected R-R events are not equally sampled date. Despite 50Hz Shimmer sampling, detected R-R values occurs in different time moments. In order to compare results in some objective method , detected R-R intervals should be interpolated (e.g spline) and resampled. This allow to be compared with measured acceleration energy - which should be taken as modulus of acceleration.

(Writer’s Answer of 8)

A. I didn't expect the part you mentioned. In this study, we just focused on the RR Interval detection without using any interpolation from the original signal of the Shimmer 3 device. If I have a chance, I would like to review and study according to your comments.

[9]- suggestion - in future research gather OBD signals from the car - accelerations and brake signals can be observed. Intensity of acceleration and frequency of braking can support information about driver's mud.

(Writer’s Answer of 9)

A. I really want to do that. Previously, I tried to buy the device to measure the strength of the accelerator, but I gave up because of the high price.

My paper was improved so much for your very helpful feedback

Thank you once again.

Best regards.

Reviewer 3 Report

The paper presents analysis of heart palpitations in elderly drivers, and derives two metrics which can predict palpitations from vehicle accelerometer data, as accelerations are the main indicator that the driving is unsafe and perhaps stressful.

I like the research presented and I think it is of great importance for automobile safety in elderly drivers. However, the paper is very poorly written and I struggled with understanding, and even now I am not sure I got everything right. Therefore the paper is NOT suitable to be accepted as is. I am not sure whether even the description in the first paragraph of my review is entirely correct!

There are the following issues of the paper, first the major ones:

1) Introduction: define and describe complete physiological reasoning behind the paper, preferrably in the separate subsection. At the beginning, you talk of heart attacks, which is confusing because abstract talks about palpitations, only much later it is revealed that palpitations can be precursor to heart attack. Therefore, describe the complete physiology you rely on to reader which comes from computing or engineering background.

2) I still cannot clearly see the use of that system, indeed before I read the paper I thought I know how would I use ECG in vehicles, but now I am confused more than before. Why not analyze ECG and when palpitations are detected, alert the driver (or slow the vehicle, or whatever the safe maneouver is?) What is your intended use? Use ECG? Use estimaded SDI from car sensors and do not use ECG at all? I am confused.

3) Rewrite the Overview. Introduce all new concepts, slowly, and in logical order, so the reader can follow. In the whole article I cannot find clear definition what RR interval is. Is that an interval between two R peaks? Which of two signals you use is used to detect R peaks?

4) Why is unsupervised learning used at all? I would assume you take the clinical definition of unsafe palpitations, process the ECG, do some kind of learning how it relates to driving parameters and that's it, you can predict RR intervals from acceleration data. Why is this not the route taken?

The smaller, more detailed comments:

5) Figure 4, what is QRS amplitude, and how does it relate to volts in the y axis?

6) Algorithm 1: can you provide short, half page, focused commentary of the algorithm? I don't even know how to associate parts of the algorithm with which parts of the text.

7) I don't understand table 3 and basically many other results. Your hypothesis was that elderly people are more at risk, because they will get heart palpitations more easily. But in table 3, the RR intervals are longer for elderly people? What, they have less intensive palpitations?

8) lines 306-308: high SDI means that the driver is prone to palpitations (short RR intervals). How is the high SDI then good?? It goes against everything I understood in the paper.

So as you see, the paper needs significant amount of work to make it clear and readable. There are also some English typos. Some of your word choices ("tensioned intervals") are also suspicious although they may be ok, I cannot judge that much.

Author Response

The paper presents analysis of heart palpitations in elderly drivers, and derives two metrics which can predict palpitations from vehicle accelerometer data, as accelerations are the main indicator that the driving is unsafe and perhaps stressful.

I like the research presented and I think it is of great importance for automobile safety in elderly drivers. However, the paper is very poorly written and I struggled with understanding, and even now I am not sure I got everything right. Therefore the paper is NOT suitable to be accepted as is. I am not sure whether even the description in the first paragraph of my review is entirely correct!

(Writer’s Answer)

A.   I really appreciate your detailed review. Thank you very much.

I revised the entire document so that main concepts such as the SDI, CRT, etc. can be seen clearly according to your advice

B.   Below is an outline of this study.

I. The heart rate of the elderly had a lot of changes during driving on average.

: (Before Driving – Driving)

Youths 24.8ms / Elderly 59.42ms

I.       However, the upper gap could not explain the safe driving briefly. So, I suggested the SDI, CRT.

II.     The main goal of this study is to present the acceleration value (=SDI, Safe Driving Intensity) for safe driving considering the different heart conditions of each subject.
à In other words, a driver’s heart is okay up to the SDI value.
à It is found that the elderly had a somewhat lower SDI than young people.

III.    The CRT is estimation value of the occurring time of heart palpitations caused by stressful driving.
à In other words, CRT is physical (heart) reaction time due to external stimuli.
à It is found that the elderly have a somewhat slower CRT than young people

IV.   Unfortunately, the relationship between CRT and SDI has not been clarified in this study. More subjects will be needed in future research.

V.    Nevertheless, our results can detect the occurring threshold of heart palpitations that is acceptable for safe driving, and can use as a basis for evaluating elderly people's driving ability.

There are the following issues of the paper, first the major ones:

1) Introduction: define and describe complete physiological reasoning behind the paper, preferrably in the separate subsection. At the beginning, you talk of heart attacks, which is confusing because abstract talks about palpitations, only much later it is revealed that palpitations can be precursor to heart attack. Therefore, describe the complete physiology you rely on to reader which comes from computing or engineering background

 (Writer’s Answer of 1)

A. I have made the correction that heart palpitations due to aggressive driving are the main reason for heart attacks in the introduction chapter.

B. Much of the introduction has been revised to make it easier to understand the contents of the thesis.

2) I still cannot clearly see the use of that system, indeed before I read the paper I thought I know how would I use ECG in vehicles, but now I am confused more than before. Why not analyze ECG and when palpitations are detected, alert the driver (or slow the vehicle, or whatever the safe maneouver is?)

What is your intended use? Use ECG? Use estimaded SDI from car sensors and do not use ECG at all? I am confused.

(Writer’s Answer of 2)

A.   First of all, the main goal of this study was to extract quantitative safe driving intensity (SDI) which can be compared with each individual.

B.   This value(SDI) is expected to be used as a basis for evaluating elderly people's driving ability.

3) Rewrite the Overview. Introduce all new concepts, slowly, and in logical order, so the reader can follow. In the whole article I cannot find clear definition what RR interval is.

Is that an interval between two R peaks? Which of two signals you use is used to detect R peaks?

(Writer’s Answer of 3)

A.   The overview has been fully revised.

B.   I added the RR Intervals definition. The following terms were somewhat confusing
: RR Interval, RR Distance, R-Peak Distance

è I decided to use it as RR Interval.

4) Why is unsupervised learning used at all? I would assume you take the clinical definition of unsafe palpitations, process the ECG, do some kind of learning how it relates to driving parameters and that's it, you can predict RR intervals from acceleration data. Why is this not the route taken?

(Writer’s Answer of 4)

A.   It was a really good question. Palpitation is a medical expression but a subjective experience. The reason for using unsupervised learning was to detect the state of heart palpitation by different people. So, palpitations were detected by K-means clustering with the suggested algorithm (Eq.5) in each persons’ ECG data.

B.   As you know, our research adapted the method finding the acceleration value that caused the heart palpitation. But, I also used your suggested routine in order to verify our SDI(Figure 6 : GV-Gradient Validation). This method is simple, but if the distribution of heart populations is not steep between each acceleration (if the SDI candidate values are similar), it is difficult to determine what the value should be SDI. So I mentioned in this paper that the part is used for verification purposes only.

The smaller, more detailed comments:

5) Figure 4, what is QRS amplitude, and how does it relate to volts in the y axis?

(Writer’s Answer of 5)

A.   I drew line and added the caption again. The left axis means RR-interval. The right axis means the acceleration value.

6) Algorithm 1: can you provide short, half page, focused commentary of the algorithm? I don't even know how to associate parts of the algorithm with which parts of the text.

(Writer’s Answer of 6)

A.   I rewrote the algorithm more briefly.

7) I don't understand table 3 and basically many other results. Your hypothesis was that elderly people are more at risk, because they will get heart palpitations more easily. But in table 3, the RR intervals are longer for elderly people? What, they have less intensive palpitations?

(Writer’s Answer of 7)

A.   Generally, the elderly have lower heart rate than young people according to many studies.

B.   The main contents of Table 3 are to express the changing heart rate during driving.

C.   The elderly have twice gap compared to the young people’s ones.
: (Before Driving – Driving)

à Youths 24.8ms / Elderly 59.42ms

8) lines 306-308: high SDI means that the driver is prone to palpitations (short RR intervals). How is the high SDI then good?? It goes against everything I understood in the paper.

(Writer’s Answer of 8)

A.    A high SDI means that the driver is NOT prone to palpitations. In other words, driver’s heart is okay up to the SDI value.
(The lack of explanation has been modified in several places.)

So as you see, the paper needs significant amount of work to make it clear and readable. There are also some English typos. Some of your word choices ("tensioned intervals") are also suspicious although they may be ok, I cannot judge that much.

(Writer’s Answer of 9)

A.   Tensioned intervals indicate heart palpitations. There was chaos in the term. All were improved.  

My paper was improved so much for your opinion.

Thank you once again.

Best regards.

Round 2

Reviewer 1 Report

-

Reviewer 3 Report

I am satisfied with the changes, however, I would ask authors that the proofread the newly introduced text (sentences in yellow), before the paper is finally accepted.